# Microbial Association with Genus *Actinomyces* in Primary and Secondary Endodontic Lesions, Review

**DOI:** 10.3390/antibiotics9080433

**Published:** 2020-07-22

**Authors:** Mario Dioguardi, Cristian Quarta, Mario Alovisi, Vito Crincoli, Riccardo Aiuto, Rolando Crippa, Francesca Angiero, Enrica Laneve, Diego Sovereto, Alfredo De Lillo, Giuseppe Troiano, Lorenzo Lo Muzio

**Affiliations:** 1Department of Clinical and Experimental Medicine, University of Foggia, Via Rovelli 50, 71122 Foggia, Italy; cristian_quarta.549474@unifg.it (C.Q.); enrica.laneve@unifg.it (E.L.); diego_sovereto.546709@unifg.it (D.S.); alfredo.delillo@unifg.it (A.D.L.); giuseppe.troiano@unifg.it (G.T.); lorenzo.lomuzio@unifg.it (L.L.M.); 2Department of Surgical Sciences, Dental School, University of Turin, 10127 Turin, Italy; mario.alovisi@unito.it; 3Department of Basic Medical Sciences, Neurosciences and Sensory Organs, Division of Complex Operating Unit of Dentistry, “Aldo Moro” University of Bari, Piazza G. Cesare 11, 70124 Bari, Italy; vito.crincoli@uniba.it; 4Department of Biomedical, Surgical, and Dental Science, University of Milan, 20122 Milan, Italy; Riccardo.Aiuto@unimi.it; 5Department of Oral Pathology, Italian Stomatological Institute, 20122 Milan, Italy; rolandocrippamd@gmail.com; 6Department of Medical Sciences and Diagnostic Integrated, S. Martino Hospital, University of Genova, 16132 Genova, Italy; f.angiero@gmail.com

**Keywords:** *Actinomyces*, apical periodontitis, endodontic failure, primary endodontic infection, secondary endodontic infection

## Abstract

The main reason for root canal treatment failure is the persistence of microorganisms after therapy, or the recontamination of the root canal system due to an inadequate seal. In the mouth, *Actinomyces* spp. constitute a significant part of the normal flora, which is indicative of their ability to adhere to oral tissue and resist cleansing mechanisms, such as salivary flow. This review, performed according to the Preferred Reporting Items for Systematic Reviews and Meta-Analysis (PRISMA), aims to clarify the prevalence of microbial genera that are associated with the genus *Actinomyces* in primary and secondary endodontic infections (primary outcome), and to identify the most prevalent species of the *Actinomyces* genus in endodontic lesions (secondary outcome). A total of 11 studies were included in the qualitative and quantitative analysis, and a total of 331 samples were analyzed. Bacteria of the genus *Actinomyces* were found in 58 samples, and 46 bacterial genera were detected in association with bacteria of the genus *Actinomyces*. Bacteria of the genus *Streptococcus* and *Propionibacterium* were those most frequently associated with *Actinomyces* in the endodontic lesions considered, and *Actinomyces israelii* was the most frequently involved species.

## 1. Introduction

The main reason for the failure of root canal treatment is the persistence of microorganisms after therapy, or the subsequent contamination of the root system due to an inadequate seal (Nair, P.N., 2004). In endodontic failures, the presence of microorganisms has been reported in 35% to 100% of cases, with Cheung et al. reporting the presence of cultivable microorganisms in 66% of samples from teeth with endodontic failures [1,2].

Modern endodontic treatment aims to remove microorganisms from the infected root canal before filling. In some circumstances, bacteria survive endodontic treatment and can cause endodontic failure [3].

The main cause of apical periodontitis is the invasion of the endodontic space by infectious agents that cause infection [4]. Although chemical and physical factors are recognized to play a role in causing apical inflammation in the scientific literature, microorganisms are considered to be fundamental in the onset and the chronicization of apical periodontal diseases [5].

Microorganisms can reach the dental pulp through the dentinal tubules, leading to carious lesions, infiltrated restorations, dental trauma and lateral periodontal lesions with apical involvement (through lateral channels or the apical foramen) [6].

Torabinejad et al. have shown that contamination (*Staphylococcus epidermidis* and *Proteus vulgaris*) from the occlusal side can reach the periapical area in less than 6 weeks, in channels blocked with gutta-percha and sealer [7]. If the temporary filling is broken, the structure of the tooth is fractured before the definitive filling, or the filling is inadequate, bacteria can access the periapical tissue and cause infections; bacteria gain access to the pulp when the thickness of the dentine between the edge of the carious lesion and the pulp is 0.2 mm [8].

Dental pulp contaminated by the presence of microorganisms is related to the onset of apical periodontal disease. The shaping and simultaneous cleansing of the infected endodontic lesion reduces the bacterial load, and the subsequent root canal filling followed by restoration creates an apical and coronal seal that increases the probability of a favorable prognosis after treatment. Endodontic failure is determined, in some cases, by the persistence of microorganisms within the canals, which determines the presence of a persistent or secondary intraradicular infection [9,10].

Microorganisms can give rise to persistent infections when they survive canal cleansing and disinfection procedures, and if bacteria infect the endodontum during treatment or after to it, they can give rise to a secondary infection [11].

The microbiota of persistent and secondary infections in endodontically treated teeth differ from those in primary infections. In fact, studies using identification procedures based on phenotypes have revealed that the microbiota in persistent infections are supported by facultative anaerobic bacteria, while in primary infections, aerobic bacteria are present with facultative anaerobes [2,12].

Numerous studies have investigated the presence of microorganisms in the endodont [2,12,13]. The species frequently found in endodontically treated teeth are *Streptococci* and *Enterococci* [14,15,16]. Therefore, many studies have focused on developing effective strategies for their eradication from the root canal [17,18].

*Actinomyces* spp. are part of the flora of the oral cavity, and have the ability to adhere to the oral tissue and thereby resist cleansing mechanisms such as salivary flow. *Actinomyces* spp. play an important role in the formation of dental biofilm; in fact, it has been suggested that *Actinomyces* species contribute to the development of diseases such as caries and periodontitis [19].

Bacterial survival is closely related to their adaptability to hostile environments; an effective survival strategy is the ability to form a biofilm, which is always present in persistent infections [20]. It is difficult to distinguish whether the microorganisms that contribute to secondary infection are those left over from primary infections, or if they are new microorganisms.

Because of the physical constraints of the root canal system, obtaining a representative sample from this site is often not an easy task. This difficulty is much more pronounced in patients undergoing remission of pulp disease, where the number of microorganisms accessible in the root canal can be low, and a number of microbial cells can be lost during procedures to remove the root canal filling.

As a result, the number of sampled cells can decrease, and the prevalence of a particular species can be underestimated.

The different percentages relating to the presence of microorganisms depend on the measurement techniques used, such as PCR or culture [21,22]. The biochemical identification of bacteria is required after isolation in pure culture, although it is laborious and time-consuming. On the other hand, some isolated oral bacteria—for example, members of *Eubacterium*—are difficult to identify using morphological and biochemical methods [23], leading to the requirement of a combination of methods in order to confirm their identification biochemically; i.e., by sequencing the 16S rRNA gene.

Molecular genetic methods—in particular, PCR—have been widely used for microbial identification purposes. PCR tests are very sensitive, and can allow the identification of microbial species that are difficult to cultivate [11,24].

More information on the different bacterial associations present within the same primary and secondary endodontic lesion can help in outlining an optimal treatment strategy for eradicating the microorganisms associated with endodontic lesions.

The purpose of this review is to investigate the possible microbial associations of actinomycetes in endodontic infections. *Actinomyces* is one of the perpetrators of persistent intra and extraradicular infections, and knowledge regarding its possible microbial associations may be important in applying a suitable therapy for eradication. In addition, the persistence of infections on the external surface of the root apex, with the formation of a biofilm, often leads to the failure of antibiotic and endodontic therapies.

## 2. Materials and Methods

The following review was performed on the basis of PRISMA (Preferred Reporting Items for Systematic Reviews and Meta-Analysis) [25] indications. The methodology has already been adopted in other systematic reviews on the topic (*Actinomyces* and *Propionibacterium*) by the same authors [26,27,28]. 

The PICO question are the following: Population—patients with teeth with primary and secondary endodontic infections;Intervention—microbial associations with the genus *Actinomyces*;Control—patients with teeth that have no *Actinomyces* infections;Outcome—odds ratio of microbial genera that are found in association with the genus *Actinomyces* in primary and secondary endodontic infections.

The primary outcome of the review is to answer the following questions: Which genera of bacteria are found in association with the genus *Actinomyces* in primary and secondary endodontic infections? What is the odds ratio of microbial genera that are found in association with the genus *Actinomyces* in primary and secondary endodontic infections? Finally, which among the species of the genus *Actinomyces* has the greatest prevalence in endodontic lesions (secondary outcome)?

After an initial selection phase, in which records were identified in databases, the potentially eligible articles were qualitatively evaluated in order to investigate the role of bacteria in endodontic infections and in apical periodontitis, with particular attention being paid to the role of *Actinomyces* in endodontic infections.

### 2.1. Eligibility Criteria

Scientific studies concerning the role of bacteria in primary and secondary endodontic lesions were considered. In particular, all studies that investigated the presence of microorganisms within dental elements subject to endodontic treatment or retreatment, conducted in recent years (40 years) and published with abstracts in English, were considered potentially eligible.

We decided to choose articles published within the last 40 years because an increasing number of new bacterial species have been identified since 1980 (according to the approved lists of bacterial names in *Med. J. Aust.* 1980, 2, 3–4) [29].

The potentially eligible articles were finally subjected to a full-text analysis so as to verify their use for qualitative analysis and quantitative analysis.

The inclusion and exclusion criteria applied in the full-text analysis were the following:Studies were included if they identified both bacteria of the genus *Actinomyces* and bacteria of other genera in dental elements subjected to endodontic treatment or retreatment, or in the teeth subjected to apicectomy or extraction following endodontic failure;Studies were excluded if they did not report the prevalence data for bacteria of the genus *Actinomyces* in the primary and secondary lesions of the dental elements, did not consider the microbial composition of each analyzed sample, tested the presence of only a few species of bacteria, were not written in English or were published before 1980.

### 2.2. Research Methodology

The articles were identified using electronic databases—namely PubMed and Scopus—and their bibliographies were examined and consulted in order to further identify articles.

The search for sources was conducted between 13.03.2020 and 25.03.2020.

The following search terms were used in the searches of PubMed, Scopus, EBSCO and Web of Science: “persistent intraradicular infection” OR “primary endodontic infection” (PubMed 37), “endodontic failure” OR “endodontic microbiologic” (PubMed 203), “*Actinomyces*” AND “endodontic” OR “apical parodontitis” (PubMed 117), “persistent intraradicular infection” (Scopus 23), “persistent extraradicular infection” (Scopus 18), “*Actinomyces*” AND “endodontic” (Scopus 145) “persistent extraradicular infection” (EBSCO 7), “persistent intraradicular infection” (EBSCO 14), “*Actinomyces*” AND “endodontic” (EBSCO 113), “persistent extraradicular infection” (Web of Science 19) “persistent intraradicular infection” (Web of Science 19) and “*Actinomyces*” AND “endodontic” (Web of Science 117) (Table 1). As a complement to this search, we conducted a manual evaluation of the articles included in the references of the identified full-text publications, and 51 citations were considered to be of relevance.

### 2.3. Screening Methodology

Before the identification phase of records, the keywords to be searched and their combinations were first agreed upon by the two reviewers (with the task of selecting potentially eligible articles). The records obtained were subsequently examined by two independent reviewers (M.D. and C.Q.), and a third reviewer (G.T.) acted as a decision-maker in situations of doubt.

The screening included the analysis of the title and the abstract and, in cases of doubt, a text analysis to eliminate records that were not related to the topics of the review. The articles obtained were subjected to full-text analysis by the two reviewers (81 articles), from which those eligible for qualitative analysis and inclusion in the meta-analysis for the two outcomes were identified.

The results sought by the two reviewers were the following:(1)Primary outcome—which genera of bacteria are found in association with the genus *Actinomyces* in primary and secondary endodontic infections? What is the odds ratio of microbial genera that are found in association with the genus *Actinomyces* in primary and secondary endodontic infections?(2)Secondary outcome—the determination of the prevalence of the species of the genus *Actinomyces* that has the greatest prevalence in endodontic lesions.

### 2.4. Statistical Analysis Protocol

A meta-analysis was conducted on five sub-groups identified among the genus bacteria that had the highest number of positive samples together with the genus *Actinomyces* (primary outcome). The analyzed sub-groups were the following: Streptococci, Propionibacterium, Peptostreptococci, Staphylococci and Eubacterium. With the meta-analysis of the sub-groups, odds ratios (OR) were calculated to establish whether the bacteria of the respective sub-groups were more likely to present themselves in the samples with *Actinomyces* than in those without *Actinomyces*.

The protocol with which the meta-analysis was performed is based on the indications of the Cochrane Handbook for Systematic Reviews of Interventions. It was decided to use Reviewer Manager 5.4 (Cochrane collaboration, Copenhagen, Denmark) as a software for metanalysis [30]. In particular, pooled odds ratios (OR) and their 95% confidence intervals were calculated, and the inverse of variance test was applied to test for differences in overall effects between groups. The presence of heterogeneity was assessed by calculating the Higgins Index (*I*^2^); if the measure proved to be higher than 50%, the rate of heterogeneity was considered to be high. The pooled results of meta-analysis were represented via forest plots for each of the analyzed sub-groups.

The risk of bias in the studies was calculated following the guidelines reported in the Newcastle–Ottawa Scale (NOS) for assessing the quality of studies in meta-analyses [31].

The risk of bias between studies was assessed graphically through the use of funnel plots and the calculation of heterogeneity determined through the Rev-manager 5.4 software.

A meta-regression was conducted with the use of Open Meta-Analyst version 10 (Tufts University, Medford, MA, USA) for those sub-groups that had high heterogeneity, reporting the risk of bias as a covariant.

## 3. Results

From searches in the PubMed, Scopus, EBSCO and Web of Science databases, 883 records were identified; furthermore, 51 articles included in the references of the identified full-text publications were selected. With the use of EndNote software, the overlaps were removed, resulting in 475 records. After the elimination of articles prior to 1980, 462 records remained. With the application of the eligibility criteria (all studies that studied the presence of bacteria in endodontic infection), we retained 81 articles.

Applying the inclusion and exclusion criteria, we retained 11 articles in the meta-analysis.

All articles were analyzed according to the primary and secondary outcomes as defined above.

All selection and screening procedures are described in the flowchart shown in Figure 1.

### 3.1. Study Characteristics and Data Extraction

The studies included for the quantitative analysis were those of Sunde et al., 2002 [32]; Siqueira et al., 2004 [33]; Ledezma-Rasillo et al., 2010 [34]; Sundqvist et al., 1989 [35]; Abou-Rass et al., 1998 [36]; Niazi et al., 2010 [37]; Fujii et al., 2009 [38]; Pinheiro et al., 2003 [14]; Sjogren et al., 1997 [3]; Fukushima et al., 1990 [39]; and Debelian et al., 1995 [40].

The extraction of the data and the methods by which they have been reported follow the indications of the Cochrane Handbook for Systematic Reviews of Interventions, chapter 7 (selection of studies and data collection); specifically, from pages 156 to 182.

The extracted data included the bacterium species in the infection along with the bacterial species of the genus *Actinomyces* investigated (genus and species), the article information (data, author and journal), the number of samples examined, the types of samples (tooth in pulpitis or apical periodontitis, necrotic or vital tooth, tooth previously treated endodontically, endodontic canal, and tooth with failure subject to extraction or endodontic surgery), the number of samples for pathology in the presence of *Actinomyces*, and the bacterium identification method (culture or PCR).

The data extracted for the two outcomes are shown in Table 2 and Table 3.

Table 2 reports the number of samples of a particular bacterial genus found in association with *Actinomyces*, compared with the number of samples of *Actinomyces*. Then, the number of samples with that particular genus, compared with all the samples analyzed for each article, is reported. Table 3 reports the number of samples in which each *Actinomyces* species is present in each article.

For the studies selected for qualitative and quantitative analysis, a total of 331 samples were analyzed, and bacteria of the genus *Actinomyces* were found in 58 samples. For each sample, the microbial composition was available.

For the primary outcome, the bacterial genera present in the infections were considered together with species of the genus *Actinomyces*, and the prevalence relative to the infected samples together with *Actinomyces* was calculated in addition to the absolute prevalence relative to all the samples analyzed in each study (Table 4).

In some studies, only a cultural search of bacterial species was carried out; thus, an analysis by sub-group (cultures and PCR) was also carried out to remedy an evident limit of the review, as shown in Table 5.

For the secondary outcome, the prevalence of each individual species of *Actinomyces* was calculated and compared with the total number of analyzed samples (Table 6).

### 3.2. Risk of Bias

The risk of bias was assessed using the Newcastle–Ottawa case-control scale, modified by the authors to adapt it to microbiological studies, as already done in previous systematic reviews with meta-analyses [26,27]. The results are reported in detail in Table 7. For each category, a value of one to three was assigned (where one = low and three = high).

Studies presenting a high risk of bias were not included in the meta-analysis. Articles with a high bias risk were excluded from the scale and eliminated during the inclusion phase. Other articles were excluded because for the outcomes investigated; they presented the same data and samples.

The bias risk assessment of the 11 articles included was conducted by the first reviewer (M.D.).

The risk of bias between the studies was considered low for five sub-groups of the primary research outcome; in fact, the heterogeneity that emerges from the meta-analysis shows an *I*^2^ equal to 54% for sub-group 2 (Propionibacterium), 44% for sub-group 3 (Peptostreptococcus), 30% for sub-group 4 (Staphylococcus) and 0% for sub-groups 1 and 2 (Streptococcus, Eubacterium). The low heterogeneity is also confirmed by the funnel plot (Figure 2, Figure 3, Figure 4, Figure 5 and Figure 6).

For the second sub-group, graphical analysis of the funnel plot indicates the studies of Fukushima et al. 1990 [39] and Sunde et al. 2002 [32] as possible sources of heterogeneity and bias.

Graphic evaluation of the confidence intervals for the individual studies (forest plot) shows a good overlap for the Streptococci and Eubacterium sub-groups, and poor overlap for the Propionibacterium group, confirming the lack of heterogeneity in the Streptococci and Eubacterium sub-groups, and the high heterogeneity for Propionibacterium (Cochrane Handbook for Systematic Reviews of Interventions, chapter 9.5.2, identifying and measuring heterogeneity). Since heterogeneity is a sign of a possible risk of bias between the studies, it was decided to investigate meta-regression as a function of the risk of bias determined for each individual studies.

### 3.3. Meta-Analysis

The statistical analysis of the data was performed using the Rev-manager 5.4 software (Copenhagen, 153 Denmark, The Nordic Cochrane Centre, The Nordic Cochrane Collaboration, 2014).

The meta-analysis of the first sub-group (Streptococci) showed an absence of heterogeneity with *I*^2^ equal to 0%, and a fixed effects model was applied. The results shown in Figure 7 show that Streptococci are more likely to occur in samples with *Actinomyces* (OR = 2.49; 95% confidence interval (CI): [1.27, 4.86]).

The meta-analysis of the second sub-group (Propionibacterium) showed high heterogeneity, with *I*^2^ equal to 56%, and a random effects model was applied. The results shown in Figure 8 show that Propionibacterium are not more likely to occur in samples with *Actinomyces* (OR = 1.26, 95% CI: [0.31, 5.13]).

With the identification and elimination of the two sources of heterogeneity, it is evident that *I*^2^ drops to values equal to 0%; despite the elimination of the two studies, the forest plot does not report data with statistically insignificant odds ratios in favor of the samples with *Actinomyces*.

Furthermore, a meta-regression was conducted as a function of the risk of bias evaluation within the studies, in order to investigate whether the risk of bias within the studies could be a source of heterogeneity and bias between the studies. From the statistical analysis, we find a regression coefficient equal to −0.140, with a *p* value 0.639 (Table 8). The meta-regression data are not statistically significant, and the high heterogeneity index does not depend on the bias in the studies (Figure 9).

The meta-analysis of the third sub-group (Peptostreptococci) showed average heterogeneity, with *I*^2^ equal to 44%, and a fixed effects model was applied. The results shown in Figure 10 show that Peptostreptococci are more likely to occur in samples with *Actinomyces* (OR = 2.14, 95% CI: [1.1, 4.11]).

The meta-analysis of the fourth sub-group (Staphylococci) showed average heterogeneity, with *I*^2^ equal to 30%, and a fixed effects model was applied. The results shown in Figure 11 show that Staphylococci are not more likely to occur in samples with *Actinomyces* (OR = 1.54, 95% CI: [0.54, 4.37]).

The meta-analysis of the fifth sub-group Eubacterium showed an absence of heterogeneity, with *I*^2^ equal to 0%, and a fixed effects model was applied. The results shown in Figure 12 show that Eubacterium are more likely to occur in samples with *Actinomyces* (OR = 2.68, 95% CI: [1.10, 6.51]).

## 4. Discussion

Follow-up studies report success rates of around 80–90% when canals are treated endodontically in aseptic conditions [5,41,42]. Endodontic failures mainly manifest when procedures are used that have not fulfilled the standard conditions for the elimination of microorganisms inside the endodontic lesion. Long-term follow-ups have demonstrated the presence of endodontic failures, with the presence of apical radiolucent lesions, even on teeth apparently treated adequately with procedures that meet high standards, demonstrating the persistence of infections that affect the apical portion of the dental roots.

Factors that may contribute to the perpetuation of periapical radio transparencies after root canal treatment include the following: an intraradicular infection that persists in the apical part of the root canal [42]; an extraradicular infection, generally in the form of periapical actinomycosis [42]; the filling of the extruded root canal, or other materials that cause reactions to foreign bodies [43,44,45]; and cysts, especially those with a significant accumulation of cholesterol crystals [46,47].

The samples considered in this review are primary and secondary endodontic lesions; of the 331 analyzed samples, 46 bacterial genera were detected in association with bacteria of the genus *Actinomyces*, and of these, the most frequently identified were *Streptococcus* (26/58), *Propionibacterium* (20/58), *Peptostreptococcus* (14/58), *Staphylococcus* (14/58), *Eubacterium* (13/58), *Prevotella* (10/58), *Veillonella* (8/58), *Gemella* (8/58), *Clostridium* (7/58), *Bifidobacterium* (7/58), *Lactobacillus* (7/58) and *Fusobacterium* (7/58). *Streptococcus* is the genus most frequently identified in multiple studies (eight articles), followed by *Peptostreptococcus* and *Propionibacterium* (seven articles) and then *Staphylococcus*, *Eubacterium*, *Fusobacterium* and *Prevotella* (five articles).

Other studies in the literature have examined biofilm formation in the canal space or on the outer surface of the apical portion of the root [13,48].

However, information on extraradicular infections resulting in a persistent lesion is limited, and mainly references *Actinomyces* or *Propionibacterium* species [32,33,49,50]. Bacteria that are difficult to grow are often only cultured through non-traditional methods, which leads to an underestimation of the bacterial diversity associated with persistent disease [51].

Most bacteria isolated from infected root canals are oxygen-sensitive, and cannot be grown using conventional bacteriological methods [52].

In previous studies that assessed the influence of infection on the treatment outcome, bacteriological techniques that were unfavorable for use in the recovery of anaerobic bacteria were used [53,54]. Therefore, the presence of bacteria that may have been important for the outcome of the treatment may have been precluded, and cases that apparently did not contain bacteria could, in fact, have hosted persistent microorganisms.

As PCR can overcome some of the intrinsic limitations of the culture process, it has contributed significantly to our understanding of the endodontic microbiota associated with primary infections [11].

The lesions analyzed in this review were both primary and secondary endodontic lesions; the bacteria were identified in the lesions by culture and PCR.

A study that identified multiple bacterial genera associated with *Actinomyces* was conducted by Niazi et al. (2010), who identified 35 bacterial genera in refractory endodontic lesions (9 with abscesses and 11 without abscesses) and 11 species of *Actinomyces* [37]. A considerable number of bacterial genera were also identified by Pinheiro et al. (2003) and Debelian et al. (1995), who identified 12 bacterial genera [14,40].

When using culture techniques, the microbes most commonly found in the endodontic canals of teeth with post-treatment endodontic disease are primarily Gram-positive, including rods (e.g., *Actinomyces* and *Propionibacterium*) and cocci (e.g., *Enterococcus* spp., *Streptococcus* spp.) [2,12,14,55].

The meta-analysis of the sub-groups of the five bacteria most commonly found in the samples with *Actinomyces* highlighted how the bacteria of the genera Streptococci, Peptostreptococci and Eubacterium are more likely to be found in samples positive for *Actinomyces*, compared to the negative samples, with odds ratios of 2.49, 2.14 and 2.68, respectively. The meta-analysis of the Propionibacterium and Staphylococci sub-groups indicated how testing positive for these two genera of bacteria is found with the same propensity in samples that are positive, as those which are negative, for “*Actinomyces*”.

For all the other bacteria, there are no indications from this meta-analysis suggesting their greater frequency in primary and secondary lesions with *Actinomyces*; in fact, the literature review shows us that many bacteria are more frequently found in endodontic lesions than in *Actinomyces*.

Bacteria of the genus *Actinomyces* are constantly being reclassified with the identification of new bacterial species. Bacteria that are present in the oral cavity, and that can potentially cause secondary or persistent infections, are commonly found in the gastroenteric system or in the mucous membranes of the urogenital tract [56,57].

The species of *Actinomyces* most involved in endodontic lesions are *A. israelii* (15/331), *A. naeslundii* (11/331), *A. species* (10/331), *A. viscosus* (7/331) and *A. odontolyticus* (5/331). *A. israelii* and *A. naeslundii* were identified in six different articles, *A. viscosus*, *A. meyeri* and *A. odontolyticus* were reported in three articles, and *A. radicidentis* were identified in two articles. The remaining species were identified in different individual studies.

A group of these bacteria has been shown to cause actinomycosis, progressing chronically as diseases manifesting abscesses associated with tissue fibrosis and draining sinuses, and sometimes mimicking malignant tumors [58,59,60].

*Actinomyces israelii* is the most commonly isolated species in human actinomycosis [61,62]. The loss of the integrity of the mucous membrane of the oral cavity, caused by extractions, bone and dental fractures, anesthesia, periodontal disease and the endodontic treatment of pulp exposures, can give rise to infection by these microorganisms, which, upon the interruption of the continuity of oral tissue, infect and invade the underlying tissues, thanks also to the selective conditions of anaerobiosis [63,64,65].

The limits of the study are the heterogeneity of the outcomes sought from the clinical studies included in the meta-analysis, sometimes with different microbiological identification methods, and the continuously updated taxonomy of bacterial species. The data are therefore to be considered as an indication (with an analytical basis) of which might be main bacteria associated with *Actinomyces*, which we consider to be one of the main culprits of perpetrators intraradicular and extraradicular infections.

## 5. Conclusions

Bacteria of the genera *Streptococcus* and *Propionibacterium* are those that are most frequently associated with *Actinomyces* in the considered endodontic lesions, and *Actinomyces israelii* is the most frequently involved species.

The microorganisms found in endodontic failures remain in the root canal after previous treatment, or enter during or after treatment through a leak. For all the other bacteria, the literature review shows us that many bacteria are more frequently found in endodontic lesions than in *Actinomyces*. therefore, thorough knowledge and a deep understanding of these endodontic microbes can assist in making decisions for further surgical treatment or reprocessing.

## Figures and Tables

**Figure 1 antibiotics-09-00433-f001:**
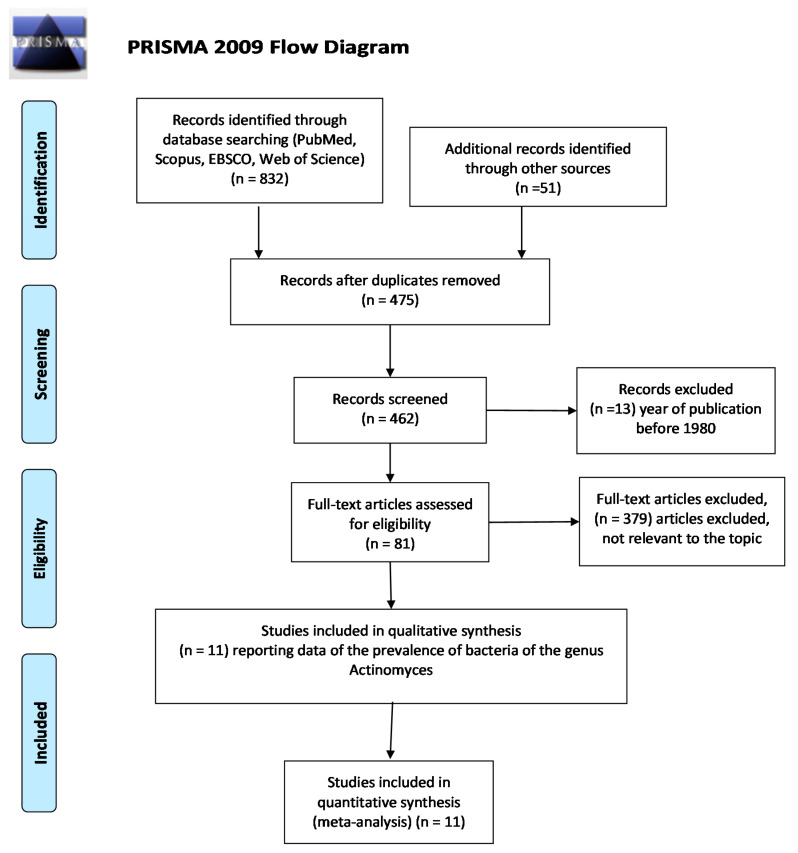
Flowchart of the different phases of the systematic review.

**Figure 2 antibiotics-09-00433-f002:**
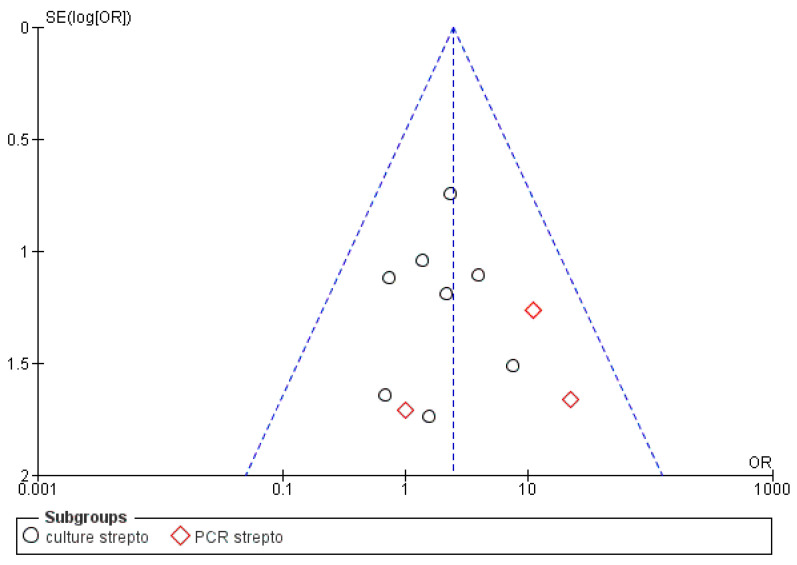
Funnel plot for the sub-group Streptococcus, *I*^2^ = 0%. The absence of heterogeneity is evident graphically. Sub-group culture *I*^2^ = 0%. Sub-group PCR *I*^2^ = 0%. The heterogeneity between the two sub-groups is *I*^2^ = 58.2%. OR: odds ratio, SE: standard error.

**Figure 3 antibiotics-09-00433-f003:**
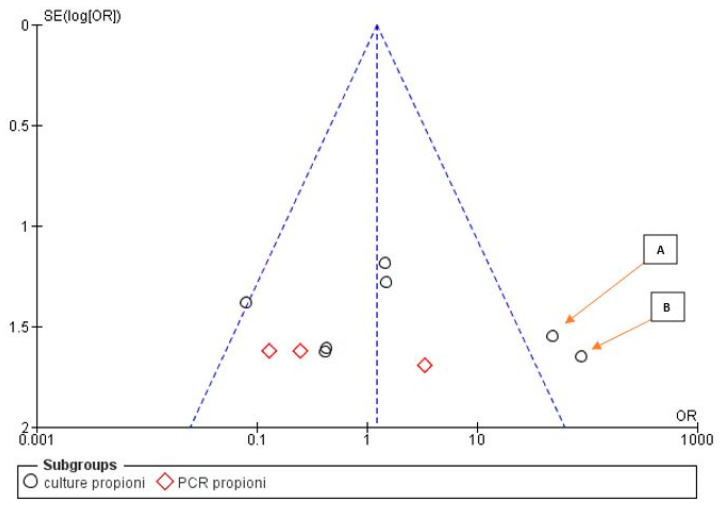
Funnel plot for the sub-group Propionibacterium, *I*^2^ = 56%. The presence of heterogeneity is highlighted graphically. The arrows highlight the sources of heterogeneity A: Fukushima, 1990; B: Sunde, 2002. Sub-group culture *I*^2^ = 64%. Sub-group PCR *I*^2^ = 5%. The heterogeneity between the two sub-groups is *I*^2^ = 13%.

**Figure 4 antibiotics-09-00433-f004:**
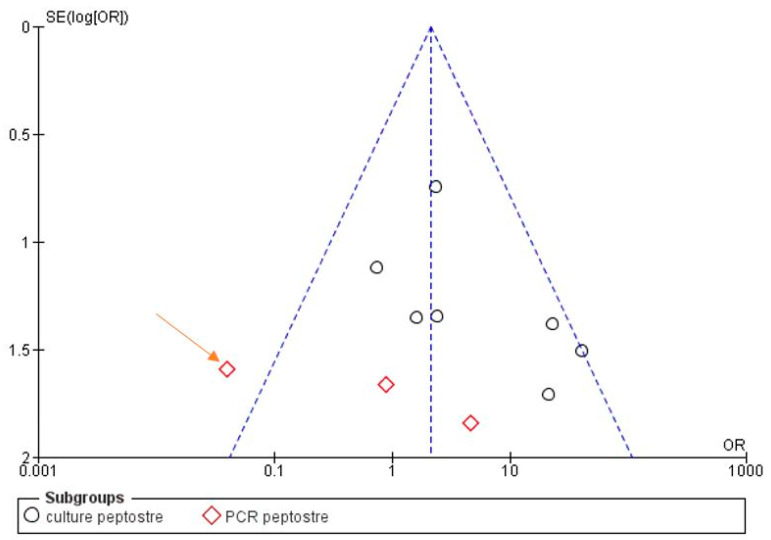
Funnel plot for the sub-group Peptostreptococcus, *I*^2^ = 44%. Sub-group culture *I*^2^ = 27%. Sub-group PCR *I*^2^ = 53%. The heterogeneity between the two sub-groups is *I*^2^ = 86.9%. The arrow indicates that the study of Niazi, 2010 is the likely source of heterogeneity.

**Figure 5 antibiotics-09-00433-f005:**
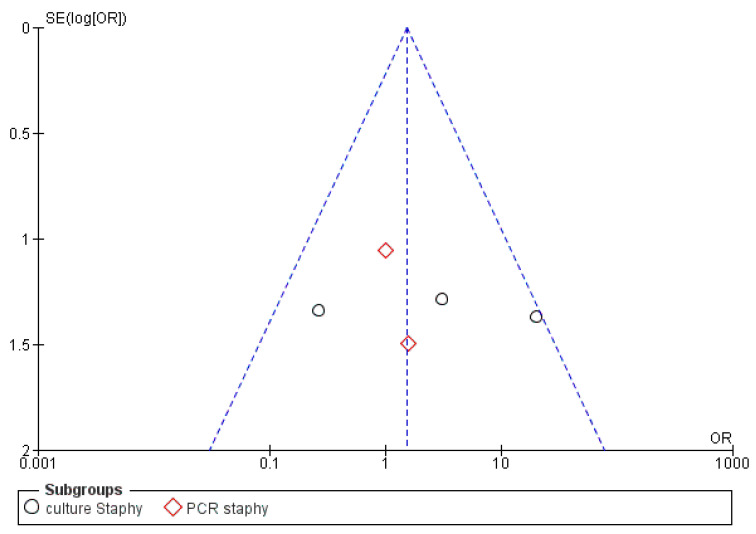
Funnel plot for the sub-group Staphylococcus, *I*^2^ = 30%. Sub-group culture *I*^2^ = 62%. Sub-group PCR *I*^2^ = 0%. The heterogeneity between the two sub-groups is *I*^2^ = 0%.

**Figure 6 antibiotics-09-00433-f006:**
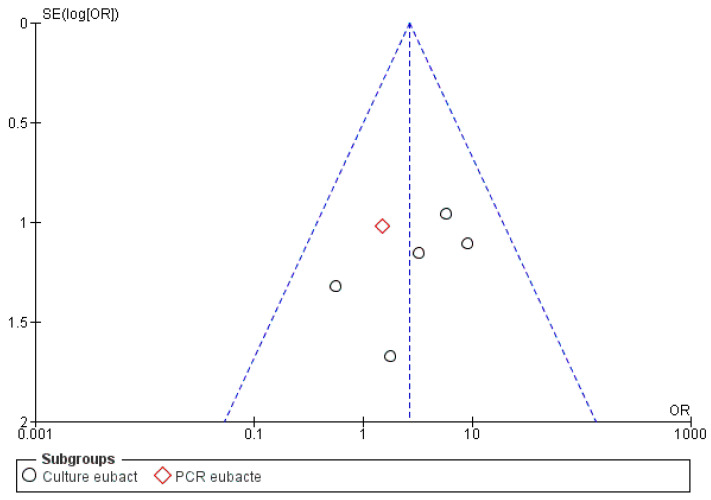
Funnel plot for the sub-group Eubacterium, *I*^2^ = 0%. Sub-group culture *I*^2^ = 0%. The heterogeneity between the two sub-groups is *I*^2^ = 0%.

**Figure 7 antibiotics-09-00433-f007:**
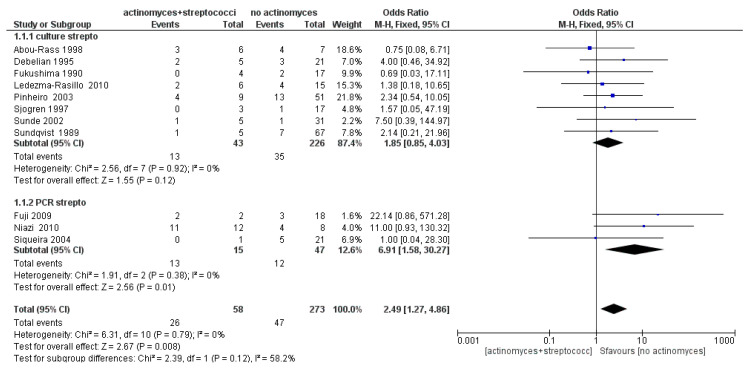
Forest plot of the fixed effects model of the meta-analysis of the sub-group Streptococci. Sub-group culture (OR = 1.85, 95% confidence interval (CI): [0.27, 4.03]), sub-group PCR (OR = 6.91, 95% CI: [1.58, 30.27]).

**Figure 8 antibiotics-09-00433-f008:**
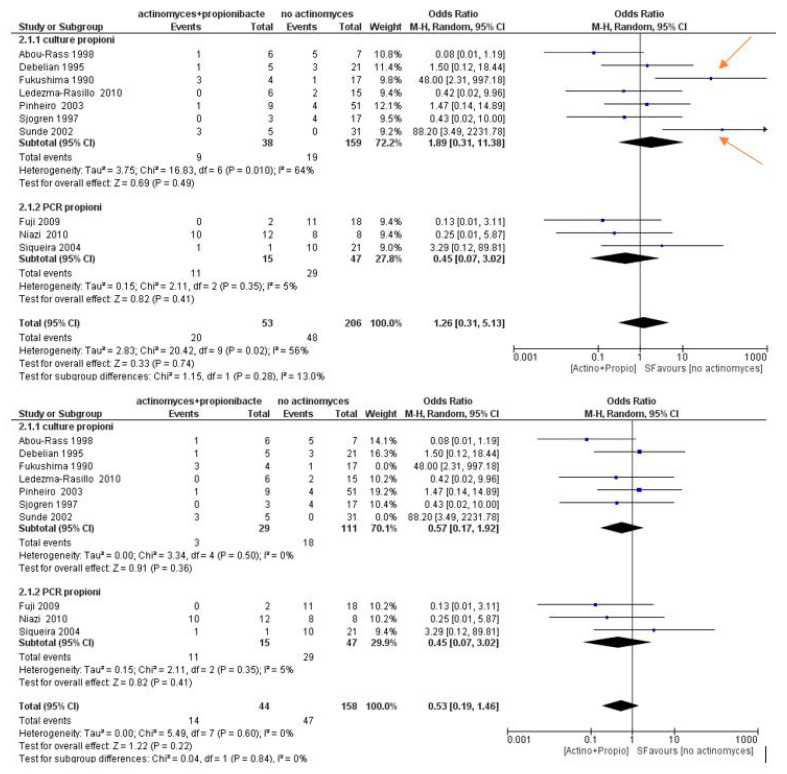
Forest plot of the random effects model of the meta-analysis of the sub-group Propionibacterium; the arrows indicate the sources of heterogeneity that are identified by the funnel plot and are also evident on the forest plot. Sub-group culture (OR = 1.89, 95% CI: [0.31, 11.38]), sub-group PCR (OR = 0.45, 95% CI: [0.07, 3.02]).

**Figure 9 antibiotics-09-00433-f009:**
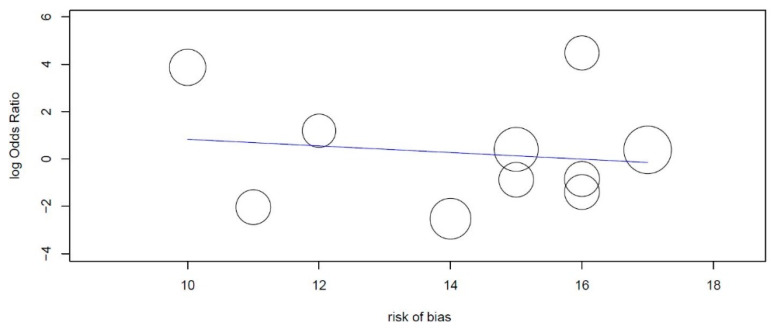
Meta-regression plot; it can be seen that the log odds ratio decreases when the risk of bias decreases, with a regression coefficient equal to −0.140 per score of risk of bias points, and with a *p* value of 0.639.

**Figure 10 antibiotics-09-00433-f010:**
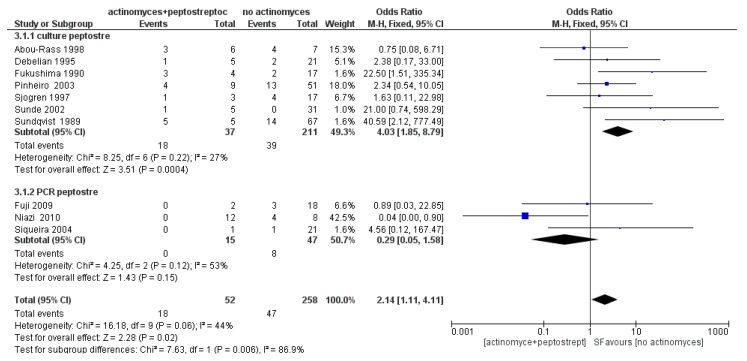
Forest plot of the fixed effects model of the meta-analysis of sub-group Peptostreptococci. Sub-group culture (OR = 4.03, 95% CI: [1.85, 8.79]), sub-group PCR (OR = 0.29, 95% CI: [0.05, 1.58]).

**Figure 11 antibiotics-09-00433-f011:**
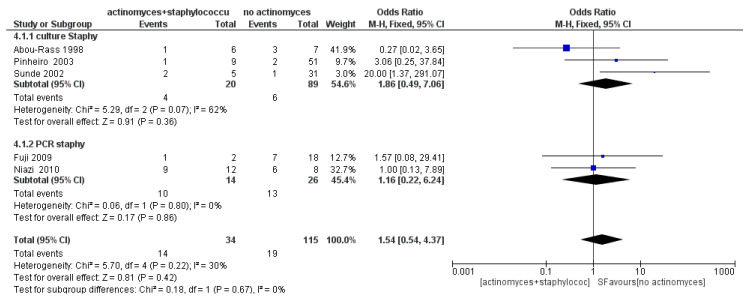
Forest plot of the fixed effects model of the meta-analysis of the sub-group Staphylococci. Sub-group culture (OR = 1.86, 95% CI: [0.49, 7.06]), sub-group PCR (OR = 1.16, 95% CI: [0.22, 6.24]).

**Figure 12 antibiotics-09-00433-f012:**
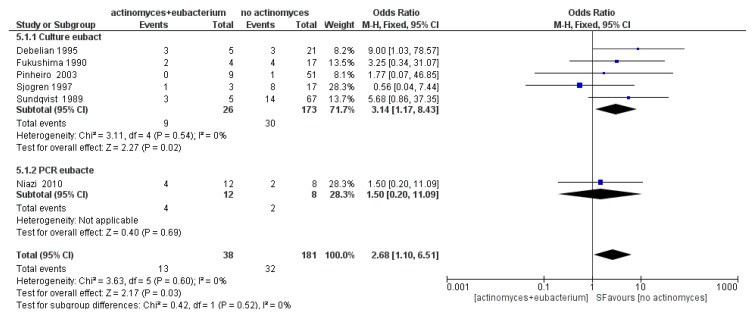
Forest plot of the fixed effects model of the meta-analysis of the sub-group Eubacterium. Sub-group culture (OR = 3.14, 95% CI: [1.17, 8.43]), sub-group PCR (OR = 1.50, 95% CI: [0.20, 11.09]).

**Table 1 antibiotics-09-00433-t001:** Complete overview of the search methodology. The overlaps were removed using EndNote 8 software. Records identified by databases: 883; records selected for quantitative analysis: 11.

Provider Database	Keywords	Search Details	No. of Records	Articles after Removal of Overlapping Articles	Number of Records after Restriction by Year of Publication (Last 40 Years)	Numbers of Articles That Have Investigated the Role of Bacteria in Endodontic Infections	Number of Studies That Consider the Microbial Composition of Each Analyzed Sample
PubMed	“persistent intraradicular infection” OR “primary endodontic infection”	“persistent intraradicular infection” [All Fields] OR “primary endodontic infection” [All Fields]	37				
PubMed	“endodontic failure” OR “endodontic microbiologic”	“endodontic failure” [All Fields] OR (endodontic [All Fields] AND microbiologic [All Fields])	203				
PubMed	“*Actinomyces*” AND “endodontic” OR “apical parodontitis”	“*Actinomyces*” [All Fields] AND “endodontic” [All Fields] OR “apical parodontitis” [All Fields]	117				
Scopus	“persistent intraradicular infection”	TITLE-ABS-KEY (persistent AND interradicular AND infection)	23				
Scopus	“persistent extraradicular infection”	TITLE-ABS-KEY (persistent AND extravascular AND infection)	18				
Scopus	“*Actinomyces*” AND “endodontic”	TITLE-ABS-KEY (“*Actinomyces*” AND “endodontic”)	145				
EBSCO	persistent extraradicular infection		7				
EBSCO	persistent intraradicular infection		14				
EBSCO	“*Actinomyces*” AND “endodontic”		113				
Web of science	persistent extraradicular infection		19				
Web of science	persistent intraradicular infection		19				
Web of science	“*Actinomyces*” AND “endodontic”		117				
Articles included in the references of the identified full-text publications			51				
Total records			883	475	462	81	11

**Table 2 antibiotics-09-00433-t002:** Primary outcome: prevalence of microbial genera in the samples in association with bacteria of the genus *Actinomyces*, and their prevalence in the total samples analyzed in each article.

First Author, Date, Journal	Type of Endodontic Lesion	Total Number of Samples	Number of Samples with *Actinomyces*	Prevalence of Microbial Genera in Association with Genus *Actinomyces*	Prevalence of Microbial Genera in the Total Samples Analyzed	Identification Method
Sunde, 2002, Journal of Endodontics	Refractory apical periodontitis	36	5	*Clostridium*: 2/5	*Clostridium: 2/36*	Culture
				*Propionibacterium: 3/5*	*Propionibacterium: 3/36*	
				*Gemella: 1/5*	*Gemella: 2/36*	
				*Peptostreptococcus: 1/5*	*Peptostreptococcus: 1/36*	
				*Vibrio: 1/5*	*Vibrio: 1/36*	
				*Leptotrichia: 1/5*	*Leptotrichia: 1/36*	
				*Staphylococcus: 2/5*	*Staphylococcus: 3/36*	
				*Streptococcus: 1/5*	*Streptococcus: 2/36*	
Siqueira, 2004, Oral surgery, oral medicine, oral pathology, oral radiology and endodontics	Root-filled teeth with persistent periradicular lesions	22	1	*Propionibacterium, Pseudoramibacter, Enterococcus*	*Propionibacterium: 11/22* *Pseudoramibacter: 12/22* *Enterococcus: 17/22*	PCR
Ledezma-Rasillo, 2010, The Journal of clinical pediatric dentistry	Primary teeth with necrotic pulps	21	6	*Bifidobacterium: 5/6*	*Bifidobacterium: 17/21*	Culture
				*Veillonella: 1/6*	*Veillonella: 2/21*	
				*Clostridium: 3/6*	*Clostridium: 7/21*	
				*Streptococcus: 2/6*	*Streptococcus: 6/21*	
				*Gemella: 1/6*	*Gemella: 1/21*	
Sundqvist, 1989, Journal of endodontics	Teeth with apical periodontitis	72	5	*Peptostreptococcus: 5/5*	*Peptostreptococcus: 19/72*	Culture
				*Lactobacillus: 2/5*	*Lactobacillus: 12/72*	
				*Bacteroides: 5/5*	*Bacteroides: 22/72*	
				*Wolinella: 1/5*	*Wolinella: 6/72*	
				*Streptococcus: 1/5*	*Streptococcus: 8/72*	
				*Eubacterium: 3/5*	*Eubacterium: 17/72*	
				*Fusobacterium: 3/5*	*Fusobacterium: 16/72*	
Abou-Rass, 1998, International endodontic journal	Closed periapical lesions associated with refractory endodontic therapy	13	6	*Streptococcus: 3/6*	*Streptococcus:7/13*	Culture
				*Staphylococcus: 1/6*	*Staphylococcus: 4/13*	
				*Peptostreptococcus: 1/6*	*Peptostreptococcus: 1/13*	
				*Gram-negative enteric rods: 1/6*	*Gram-negative enteric rods: 1/13*	
				*Propionibacterium: 1/6*	*Propionibacterium: 6/13*	
				*Porphyromonas: 1/6*	*Porphyromonas: 1/13*	
				*Fusobacterium: 1/6*	*Fusobacterium: 1/13*	
Niazi, 2010, Journal of clinical microbiology	Refractory endodontic lesions (9 with abscesses and 11 without abscesses)	20	12	*Gemella: 3/12*	*Gemella: 5/20*	PCR
				*Propionibacterium: 10/12*	*Propionibacterium: 18/20*	
				*Staphylococcus: 9/12*	*Staphylococcus:15/20*	
				*Streptococcus: 11/12*	*Streptococcus: 15/20*	
				*Clostridium: 1/12*	*Clostridium: 2/20*	
				*Capnocytophaga: 3/12*	*Capnocytophaga: 3/20*	
				*Prevotella: 4/12*	*Prevotella: 7/20*	
				*Selenomonas: 3/12*	*Selenomonas: 3/20*	
				*Olsenella: 4/12*	*Olsenella: 5/20*	
				*Bifidobacterium: 1/12*	*Bifidobacterium: 2/20*	
				*Lactobacillus: 1/12*	*Lactobacillus: 1/20*	
				*Abiotrophia: 1/12*	*Abiotrophia: 1/20*	
				*Granulicatella: 2/12*	*Granulicatella: 2/20*	
				*Kocuria: 1/12*	*Kocuria: 1/20*	
				*Micrococcus: 1/12*	*Micrococcus: 2/20*	
				*Rothia: 2/12*	*Rothia: 2/20*	
				*Eubacterium: 4/12*	*Eubacterium: 6/20*	
				*Parvimonas: 2/12*	*Parvimonas: 2/20*	
				*Solobacterium: 2/12*	*Solobacterium: 3/20*	
				*Veillonella: 3/12*	*Veillonella: 4/20*	
				*Enterococcus: 1/12*	*Enterococcus: 3/20*	
				*Bacteroides: 1/12*	*Bacteroides: 1/20*	
				*Desulfovibrio: 1/12*	*Desulfovibrio: 1/20*	
				*Lautropia: 1/12*	*Lautropia: 1/20*	
				*Phascolarctobacterium: 1/12*	*Phascolarctobacterium: 1/20*	
				*Catonella: 1/12*	*Catonella: 1/20*	
				*Oribacterium: 1/12*	*Oribacterium: 1/20*	
				*Slackia: 2/12*	*Slackia: 4/20*	
				*Pseudoramibacter: 3/12*	*Pseudoramibacter: 4/20*	
				*Mogibacterium: 3/12*	*Mogibacterium: 6/20*	
				*Atopobium: 2/12*	*Atopobium: 2/20*	
				*Dialister: 3/12*	*Dialister: 5/20*	
				*Porphyromonas: 2/12*	*Porphyromonas:2/20*	
				*Tanerella: 1/12*	*Tanerella: 4/20*	
				*Campylobacter: 1/12*	*Campylobacter: 2/20*	
Fujii, 2009, Oral microbiology and immunology	Apical periodontitis lesions of obturated teeth	20	2	*Fusobacterium: 1/2*	*Fusobacterium: 5/20*	PCR
				*Slackia: 1/2*	*Slackia: 1/20*	
				*Staphylococcus: 1/2*	*Staphylococcus: 8/20*	
				*Streptococcus: 2/2*	*Streptococcus: 5/20*	
				*Stenotrophomonas: 1/2*	*Stenotrophomonas: 1/20*	
				*Prevotella: 1/2*	*Prevotella: 4/20*	
Pinheiro, 2003, International endodontic journal	Root-filled teeth with apical periodontitis	60	9	*Streptococcus: 4/9*	*Streptococcus: 17/60*	Culture
				*Enterococcus: 2/9*	*Enterococcus: 28/60*	
				*Prevotella: 2/9*	*Prevotella: 6/60*	
				*Peptostreptococcus: 2/9*	*Peptostreptococcus: 9/60*	
				*Bifidobacterium: 1/9*	*Bifidobacterium: 1/60*	
				*Veillonella: 3/9*	*Veillonella: 4/60*	
				*Candida: 1/9*	*Candida: 2/60*	
				*Propionibacterium: 1/9*	*Propionibacterium: 5/60*	
				*Fusobacterium: 1/9*	*Fusobacterium: 3/60*	
				*Gemella: 3/9*	*Gemella: 4/60*	
				*Haemophilus: 1/9*	*Haemophilus: 1/60*	
				*Staphylococcus: 1/9*	*Staphylococcus: 3/60*	
Sjogren, 1997, International endodontic journal	Apical periodontitis	20	3	*Prevotella: 1/3*	*Prevotella: 3/20*	Culture
				*Eubacterium: 1/3*	*Eubacterium: 9/20*	
				*Campylobacter: 1/3*	*Campylobacter: 4/20*	
				*Peptostreptococcus: 1/3*	*Peptostreptococcus: 5/20*	
Fukushima, 1990, Journal of endodontics	Untreated cases	21	4	*Propionibacterium: 3/4*	*Propionibacterium: 4/21*	Culture
				*Lactobacillus: 3/4*	*Lactobacillus: 5/21*	
				*Eubacterium: 2/4*	*Eubacterium: 6/21*	
				*Peptostreptococcus: 3/4*	*Peptostreptococcus: 5/21*	
				*Peptococcus: 1/4*	*Peptococcus: 2/21*	
Debelian et al., 1995, Endodontics & Dental Traumatology	Teeth with asymptomatic apical periodontitis	26	5	*Propionibacterium: 1/5*	*Propionibacterium: 4/26*	Culture
				*Prevotella: 2/5*	*Prevotella: 5/26*	
				*Eubacterium: 3/5*	*Eubacterium: 6/26*	
				*Campylobacter: 1/5*	*Campylobacter: 1/26*	
				*Veillonella: 1/5*	*Veillonella: 2/26*	
				*Lactobacillus: 1/5*	*Lactobacillus: 1/26*	
				*Streptococcus: 2/5*	*Streptococcus: 5/26*	
				*Porphyromonas: 1/5*	*Porphyromonas: 2/26*	
				*Fusobacterium: 1/5*	*Fusobacterium: 4/26*	
				*Clostridium: 1/5*	*Clostridium: 1/26*	
				*Peptostreptococcus: 1/5*	*Peptostreptococcus: 3/26*	
				*Saccharomyces: 1/5*	*Saccharomyces: 1/26*	

**Table 3 antibiotics-09-00433-t003:** Secondary outcome (prevalence of species of the genus *Actinomyces* given the total number of samples analyzed in each article).

First Author, Date, Journal	Type of Endodontic Lesion	Total Number of Samples	Number of Samples with *Actinomyces*	Prevalence of Individual Species of the Genus *Actinomyces*, Given the Total Number of Analyzed Samples	Identification Method
Sunde, 2002, Journal of endodontics	refractory apical periodontitis	36	5	*Actinomyces israelii*: 3/36 *Actinomyces viscosus*: 2/36 *Actinomyces meyeri*: 1/36 *Actinomyces naeslundii*: 1/36	Culture
Siqueira, 2004, Oral surgery, oral medicine, oral pathology, oral radiology, and endodontics	Root-filled teeth with persistent periradicular lesions	22	1	*Actinomyces radicidentis*: 1/22	PCR
Ledezma-Rasillo, 2010, The Journal of clinical pediatric dentistry	Primary teeth with necrotic pulps	21	6	*Actinomyces israelii*: 4/21 *Actinomyces naeslundii*: 2/21	Culture
Sundqvist, 1989, Journal of endodontics	Teeth with apical periodontitis	72	5	*Actinomyces species*: 5/72	Culture
Abou-Rass, 1998, International endodontic journal	Closed periapical lesions associated with refractory endodontic therapy	13	6	*Actinomyces* sp. I: 1/13 *Actinomyces* sp. II: 1/13 *Actinomyces* sp.: 5/13	Culture
Niazi, 2010, Journal of clinical microbiology	Refractory endodontic lesions (9 with abscesses and 11 without abscesses)	20	12	Actinomyces gerencseriae oral taxon 618: 1/20 *Actinomyces* sp. *oral clone CT047*: 1/20 *Actinomyces massiliensis*: 1/20 *Actinomyces meyeri*: 1/36 *Actinomyces radicidentis*: 1/36 *Actinomyces* sp. *oral taxon 169 clone AG004*: 3/36 *Actinomyces israelii*: 1/36 *Actinomyces* sp. *oral clone JA063*: 1/36 *Actinomyces* sp. oral taxon 181 strain Hal1065: 1/36 *Actinomyces* strain B27SC: 2/36 *Actinomyces* strain C29KA: 1/36	PCR
Fujii, 2009, Oral microbiology and immunology	Apical periodontitis lesions of obturated teeth	20	2	*Actinomyces naeslundii*: 2/20	PCR
Pinheiro, 2003, International endodontic journal	Root-filled teeth with apical periodontitis	60	9	*A. naeslundii*: 4/60 *A. viscosus*: 3/60 *A. odontolyticus*: 3/60	Culture
Sjogren, 1997, International endodontic journal	Apical periodontitis	20	3	*Actinomyces israelii*: 2/20 *Actinomyces odontolyticus*: 1/20 *Actinomyces naeslundii*: 1/20	Culture
Fukushima, 1990, Journal of endodontics	Untreated cases	21	4	*Actinomyces israelii*: 2/21 *Actinomyces viscosus*: 2/21 *A. meyeri*: 1/21	Culture
Debelian et al., 1995, Endodontics & Dental Traumatology	Teeth with asymptomatic apical periodontitis	26	5	*Actinomyces israelii*: 3/26 *Actinomyces naeslundii*: 1/26 *Actinomyces odontolyticus*: 1/26	Culture

**Table 4 antibiotics-09-00433-t004:** Total prevalence of bacterial genera in association with *Actinomyces* with respect to all samples, as well as the total number of samples for all articles selected for this review. The total number of positive samples for each single bacterium is also reported.

Bacterial Genus	Prevalence in Samples That Were Associated with *Actinomyces*, Given the Total Number of Samples for All Articles Selected for This Review	Prevalence in Samples, Given the Total Number of Samples for All Articles Selected for This Review	Number of Articles Reporting This Genus
*Clostridium*	7/58	12/331(3.6%)	4
*Propionibacterium*	20/58	51/331(15.4%)	7
*Gemella*	8/58	12/331(3.6%)	4
*Peptostreptococcus*	14/58	43/331(13%)	7
*Vibrio*	1/58	1/331(0.3%)	1
*Leptotrichia*	1/58	1/331(0.3%)	1
*Staphylococcus*	14/58	33/331(10%)	5
*Streptococcus*	26/58	65/331(19.6%)	8
*Pseudoramibacter*	4/58	16/331(4.8%)	2
*Enterococcus*	4/58	48/331(14.5%)	3
*Bifidobacterium*	7/58	20/331(6%)	3
*Veillonella*	8/58	12/331(3.6%)	4
*Lactobacillus*	7/58	19/331(5.7%)	4
*Bacteroides*	6/58	23/331(6.9%)	2
*Wolinella*	1/58	6/331(1.8%)	1
*Eubacterium*	13/58	44/331(13.3%)	5
*Fusobacterium*	7/58	29/331(8:8%)	5
*Gram-negative enteric rods*	1/58	1/331(0.3%)	1
*Porphyromonas*	4/58	5/331(1.5%)	3
*Capnocytophaga*	3/58	3/331(0.9%)	1
*Prevotella*	10/58	25/331(7.5%)	5
*Selenomonas*	3/58	3/331(0.9%)	1
*Olsenella*	4/58	5/331(1.5%)	1
*Abiotrophia*	1/58	1/331(0.3%)	1
*Granulicatella*	2/58	2/331(0.6%)	1
*Kocuria*	1/58	1/331(0.3%)	1
*Micrococcus*	1/58	2/331(0.6%)	1
*Rothia*	2/58	2/331(0.6%)	1
*Parvimonas*	2/58	2/331(0.6%)	1
*Solobacterium*	2/58	3/331(0.9%)	1
*Desulfovibrio*	1/58	1/331(0.3%)	1
*Lautropia*	1/58	1/331(0.3%)	1
*Phascolarctobacterium*	1/58	1/331(0.3%)	1
*Catonella*	1/58	1/331(0.3%)	1
*Oribacterium*	1/58	1/331(0.3%)	1
*Slackia*	3/58	5/331(1.5%)	2
*Mogibacterium*	3/58	6/331(1.8%)	1
*Atopobium*	2/58	2/331(0.6%)	1
*Dialister*	3/58	5/331(1.5%)	1
*Tanerella*	1/58	4/331(1.2%)	1
*Campylobacter*	3/58	7/331(2.1%)	3
*Stenotrophomonas*	1/58	1/331(0.3%)	1
*Candida*	1/58	2/331(0.6%)	1
*Haemophilus*	1/58	1/331(0.3%)	1
*Peptococcus*	1/58	2/331(0.6%)	1
*Saccharomyces*	1/58	1/331(0.3%)	1

**Table 5 antibiotics-09-00433-t005:** Data referring to the two sub-groups (culture and PCR).

Bacterial Genus	Sub-Group Culture		Sub-Group PCR	
	Prevalence in Samples That Were Associated with *Actinomyces*	Prevalence in Samples, Given the Total of Number of Samples	Prevalence in Samples That Were Associated with *Actinomyces*	Prevalence in Samples, Given the Total of Number of Samples
*Clostridium*	6/16	10/83	1/12	2/20
*Propionibacterium*	9/29	22/156	11/13	29/42
*Streptococcus*	13/36	43/228	13/14	20/40
*Peptostreptococcus*	14/37	43/212	-	-
*Staphylococcus*	13/34	25/129	1/2	8/20
*Eubacterium*	9/17	38/139	4/12	6/20
*Fusobacterium*	6/25	24/171	1/2	5/20
*Prevotella*	9/29	21/126	1/2	4/20
*Veillonella*	5/20	8/107	3/9	4/20
*Lactobacillus*	6/14	18/119	1/12	1/20
*enterococcus*	2/9	28/60	2/13	20/42
*Porphyromonas*	2/11	3/39	2/12	2/20
*Campylobacter*	2/8	5/46	1/12	2/20
*Bifidobacterium*	6/15	18/81	1/12	2/20

**Table 6 antibiotics-09-00433-t006:** Prevalence of the individual *Actinomyces* species, given the total number of samples for all articles selected for this review. We have the greatest number of positive samples with *Actinomyces israelii* and *Actinomyces naeslundii*.

Species of the Genus *Actinomyces*	Prevalence of the *Actinomyces* Species in Samples for the Total Number of Analyzed Samples for All Articles	Number of Articles Reporting This Species
*Actinomyces israelii*	15/331(4.5%)	6
*Actinomyces viscosus*	7/331(2.1%)	3
*Actinomyces meyeri*	3/331 (0.9%)	3
*Actinomyces naeslundii*	11/331(3.3%)	6
*Actinomyces radicidentis*	2/331(0.6%)	2
*Actinomyces species*	10/331(3%)	2
*Actinomyces* sp. *I*	1/331(0.3%)	1
*Actinomyces* sp. *II*	1/331(0.3%)	1
*Actinomyces gerencseriae oral taxon 618*	1/331(0.3%)	1
*Actinomyces* sp. *oral clone CT047*	1/331(0.3%)	1
*Actinomyces massiliensis*	1/331(0.3%)	1
*Actinomyces* sp. *oral taxon 169 clone AG004*	3/331(0.9%)	1
*Actinomyces* sp. *oral clone JA063*	1/331(0.3%)	1
*Actinomyces* sp. *oral taxon 181 strain Hal1065*	1/331(0.3%)	1
*Actinomyces strain B27SC*	2/331(0.6%)	1
*Actinomyces strain C29KA*	1/331(0.3%)	1
*Actinomyces odontolyticus*	5/331(1.5%)	3

**Table 7 antibiotics-09-00433-t007:** Assessment of the risk of bias within the studies (Newcastle–Ottawa scale), with scores 7 to 12 = low quality, 13 to 20 = intermediate quality, and 21 to 24 = high quality.

		Selection			Comparability		Exposure		Score	Sub-Group
Reference	Definition of Cases	Representativeness of Cases	Selection of Controls	Definition of Controls	Comparability of Cases and Controls on the Basis of the Design or Analysis	Ascertainment of Exposure	Same Method of Ascertainment for Cases and Controls	Non-Response Rate		
*[34]* Ledezma-Rasillo et al., *2010 The Journal of clinical pediatric dentistry*	*3*	*1*	*2*	*2*	*2*	*2*	*3*	*0*	*15*	*Streptococcus, Propionibacterium,*
*[37]* Niazi et al., *2010 Journal of endodontics*	*3*	*1*	*3*	*3*	*2*	*1*	*3*	*0*	*16*	*Streptococcus, Propionibacterium, Peptostreptococcus, Staphylococcus, Eubacterium*
*[38]* Fujii et al., *2009 Oral microbiology and immunology*	*2*	*2*	*1*	*1*	*1*	*2*	*2*	*0*	*11*	*Streptococcus, Propionibacterium, Peptostreptococcus, Staphylococcus*
*[33]* Siqueira et al., *2004 Oral surgery, oral medicine, oral pathology, oral radiology, and endodontics*	*3*	*1*	*1*	*1*	*1*	*3*	*2*	*0*	*12*	*Streptococcus, Propionibacterium, Peptostreptococcus*
*[14]* Pinheiro et al., *2003 International endodontic journal*	*2*	*2*	*2*	*2*	*3*	*3*	*3*	*0*	*17*	*Streptococcus, Propionibacterium, Peptostreptococcus, Staphylococcus, Eubacterium*
*[32]* Sunde et al., *2002 Journal of endodontics*	*2*	*2*	*2*	*2*	*3*	*2*	*3*	*0*	*16*	*Streptococcus, Propionibacterium, Peptostreptococcus, Staphylococcus*
*[3]* Sjogren et al., *1997 International endodontic journal*	*2*	*2*	*2*	*2*	*3*	*2*	*3*	*0*	*16*	*Streptococcus, Propionibacterium, Peptostreptococcus, Eubacterium*
*[40]* Debelian, et al. *1995 Endodontics & dental traumatology*	*2*	*2*	*2*	*2*	*3*	*2*	*2*	*0*	*15*	*Streptococcus, Propionibacterium, Eubacterium*
*[39]* Fukushima et al., *1990 Journal of endodontics*	*2*	*1*	*2*	*1*	*1*	*1*	*2*	*0*	*10*	*Streptococcus, Propionibacterium, Peptostreptococcus, Eubacterium*
*[35]* Sundqvist et al., *1989 Journal of endodontics*	*3*	*3*	*3*	*1*	*1*	*1*	*2*	*0*	*14*	*Streptococcus, Peptostreptococcus, Eubacterium*
*[36]* Abou-Rass et al., *1998* *International endodontic journal*	*2*	*2*	*2*	*2*	*2*	*2*	*2*	*0*	*14*	*Streptococcus, Propionibacterium, Peptostreptococcus, Staphylococcus*

**Table 8 antibiotics-09-00433-t008:** Random effects model: regression results for the risk of bias.

Covariate	Coefficients	Lower Bound	Upper Bound	Std. Error	*Z*-Value	*p*-Value
Intercept	2.226	−6.222	10.675	4.310	0.5164	0.605
Risk of bias	−0.140	−0.724	0.444	0.298	−0.4697	0.639

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
