# Peer review of "Microbial Association with Genus Actinomyces in Primary and Secondary Endodontic Lesions, Review"

_antibiotics, 2020, doi:10.3390/antibiotics9080433_

Round 1
Reviewer 1 Report
The paper entitled “Microbial Association with Genus Actinomyces in Primary and Secondary Endodontic Lesions Systematic Review with Meta-Analysis” is a remarkably interesting work that evaluates the prevalence of microbial genera are associated with the genus Actinomyces in root canal infections. The authors present and exhaustive analysis by performing a Systematic Review with Meta-Analysis in several publications.
Overall, this manuscript can be considered for publication. However, I have some comments for further improvement and understanding of the manuscript, mostly in the results section.
Major comments:
Figure 1: The flowchart could be nicely improved by using free tools online https://support.microsoft.com/en-us/office/create-a-basic-flowchart-in-visio-e207d975-4a51-4bfa-a356-eeec314bd276 a good visualization is the key to understanding.
Figure 2: The quality of the figures is not optimal for reading. The funnel plots layout don’t have any numbering or letters, so it is not known what they are referring to the bacterial sub-groups. Although axis are named SE and OR it is assumed that refer to Standard errors and Odd ratios, that should be stated on the figure caption, not assumed by the reader. Instead of having the 11 articles cramped in the plot, maybe a legend and symbols would be easy to visualize.
Tables are a good way to summarize data. Overall the tables on the manuscript could be highly improved.
Column headings should NOT have words cut between lines
It would be more understandable to read percentages rather than fractions whenever can be applicable.
Table titles should provide more information and clarity.
All tables formatting (fonts, size and formatting) should be uniform throughout the manuscript.
There are several punctuation errors along the manuscript.
Author Response
The paper entitled “Microbial Association with Genus Actinomyces in Primary and Secondary Endodontic Lesions Systematic Review with Meta-Analysis” is a remarkably interesting work that evaluates the prevalence of microbial genera are associated with the genus Actinomyces in root canal infections. The authors present and exhaustive analysis by performing a Systematic Review with Meta-Analysis in several publications.
Overall, this manuscript can be considered for publication. However, I have some comments for further improvement and understanding of the manuscript, mostly in the results section.
Major comments:
Figure 1: The flowchart could be nicely improved by using free tools online https://support.microsoft.com/en-us/office/create-a-basic-flowchart-in-visio-e207d975-4a51-4bfa-a356-eeec314bd276 a good visualization is the key to understanding.
Figure 2: The quality of the figures is not optimal for reading. The funnel plots layout don’t have any numbering or letters, so it is not known what they are referring to the bacterial sub-groups. Although axis are named SE and OR it is assumed that refer to Standard errors and Odd ratios, that should be stated on the figure caption, not assumed by the reader. Instead of having the 11 articles cramped in the plot, maybe a legend and symbols would be easy to visualize.
Tables are a good way to summarize data. Overall the tables on the manuscript could be highly improved.
Column headings should NOT have words cut between lines
It would be more understandable to read percentages rather than fractions whenever can be applicable.
Table titles should provide more information and clarity.
All tables formatting (fonts, size and formatting) should be uniform throughout the manuscript.
There are several punctuation errors along the manuscript.
Answer
Thanks for the Comments and Suggestions
- the flow chart has been completely redesigned as recommended, based on the Flow chart Prisma
- the funnel plots have been divided and graphically improved with a better explanation and legend (SE and OR )
- The tables have been modified and formatted as required. the percentage values ​​have been added where applicable
- the manuscript has been revised by MDPI English Editing
Reviewer 2 Report
The manuscript required English minor spell check.
Have you submitted the data to Prospero system?
If it is yes, can you report the code in the text? If it is not, you should remove the word systematic and meta-analysis form the title. You can just leave the word “review”
Author Response
the manuscript required English minor spell check.
Have you submitted the data to Prospero system?
If it is yes, can you report the code in the text? If it is not, you should remove the word systematic and meta-analysis form the title. You can just leave the word “review”
Answer
Thanks for the Comments and Suggestions
- the manuscript has been revised by MDPI English Editing
- I removed the systematic word and meta-analysis from the title
Reviewer 3 Report
The manuscript entitled: “Microbial Association with Genus Actinomyces in Primary and Secondary Endodontic Lesions. Systematic Review with Meta-Analysis” is very interesting and addresses an issue with great repercussion in the field of endodontics. At the methodological level, it is of high quality and follows the recommendations of PRISMA and Cochrane. There are several points that as a reviewer raise doubts for me and should be clarified to increase the quality of the manuscript.
-The revision has been registered in PROSPERO?
-Can prevalence really be considered as the intervention in the PICO question?
-The component O of the PICO question should not be the OR?
-Normally bibliographic searches are usually carried out in at least 3 databases, why have you used only two? Why not use EMBASE? Justify the choice of Scopus and pubmed as databases.
-You should use the PRISMA flow chart as figure 1.
-The results should be enriched with a deeper statistical analysis of publication bias.
-The authors consider that a Higgins index I2> 50% indicates high heterogeneity. So why do they analyze the meta-analysis of the second Sub-group (Propionibacterium) using a fixed effects model? Wouldn't it be logical to use a random effects model?
-Do the OR estimates obtained vary if the studies were analyzed with PCR or culture? It would be interesting to do a subgroup analysis (PCR vs. Culture) in each of the 5 meta-analysis performed, if possible, to assess whether the OR differs.
-The study data show that bacteria exist much more frequently associated with endodontic lesions than Actinomyces. Perhaps, it should be better clarified in the discussion and in the conclusion.
For all these reasons, I consider that the manuscript requires further revision.
Author Response
Answer
Thanks for the Comments and Suggestions
-The revision has been registered in PROSPERO?
It has not been registered.
-Can prevalence really be considered as the intervention in the PICO question?
-The component O of the PICO question should not be the OR?
The pico question was changed as requested
-Normally bibliographic searches are usually carried out in at least 3 databases, why have you used only two? Why not use EMBASE? Justify the choice of Scopus and pubmed as databases.
2 other EBSCO and WEB of Science databases added to the search
-You should use the PRISMA flow chart as figure 1.
the PRISMA flow chart inserted as required
-The results should be enriched with a deeper statistical analysis of publication bias.
As requested, a more in-depth statistical analysis was performed with the execution of a metaregression based on the risk of partiality and the search for sources of heterogeneity
-The authors consider that a Higgins index I2> 50% indicates high heterogeneity. So why do they analyze the meta-analysis of the second Sub-group (Propionibacterium) using a fixed effects model? Wouldn't it be logical to use a random effects model?
It has been changed to random effects
-Do the OR estimates obtained vary if the studies were analyzed with PCR or culture? It would be interesting to do a subgroup analysis (PCR vs. Culture) in each of the 5 meta-analysis performed, if possible, to assess whether the OR differs.
Odd ratios were performed for the Culture and PCR Subgroups
-The study data show that bacteria exist much more frequently associated with endodontic lesions than Actinomyces. Perhaps, it should be better clarified in the discussion and in the conclusion.
added the following statements in the discussion and conclusions
For all the other bacteria, there are no indications from this meta-analysis showing a greater frequency in primary and secondary lesions with Actinomyces; in fact, the literature review shows us that many bacteria are more frequently found in endodontic lesions than in Actinomyces.
Round 2
Reviewer 1 Report
The authors have address the comments and suggestions. I would check once again for misspelling errors and/or bacteria names in italics.
Actinomyces for example is not in italics in:
page 6: paragraph 1
page 7: last paragraph
Fig. 1
page 13
table 5
table 6 caption
page 22
page 24
page 25
Reviewer 3 Report
The authors have made all the suggested changes and haveanswered the questions and concerns raised. Therefore, I
consider that the manuscript is acceptable for publication in the current format.